



# Technical note: sea salt interference with black carbon quantification in snow samples using the single particle soot photometer

Marco Zanatta[1,a], Andreas Herber[1], Zsófia Jurányi[1], Oliver Eppers[2,3], Johannes Schneider[2], Joshua P. Schwarz[4]

[1] Alfred-Wegener-Institut, Helmholtz-Zentrum für Polar- und Meeresforschung (AWI), Bremerhaven, Germany
[2] Particle Chemistry Department, Max Planck Institute for Chemistry, Mainz, Germany
[3] Johannes Gutenberg University of Mainz, Institute for Atmospheric Physics, Mainz, Germany
[4] Chemical Sciences Laboratory, Earth System Research Laboratories, Boulder, CO, USA
[a] now at LISA, UMR CNRS 7583, Université Paris-Est-Créteil, Université de Paris, Institut Pierre Simon Laplace (IPSL), Créteil, France

*Correspondence to*: Andreas Herber (andreas.herber@awi.de)

## Abstract

After deposition from the atmosphere, black carbon aerosol (BC) takes part in the snow albedo feedback contributing to modification of the Arctic radiative budget. With the initial goal of quantifying the concentration of BC in the Arctic snow and subsequent climatic impacts, snow samples were collected during the Polarstern expedition PASCAL (Polarstern cruise 106) in the sea ice covered Fram Strait in early summer 2017. The content of refractory BC (rBC) was then quantified in the laboratory of the Alfred Wegener Institute with the single particles soot photometer (SP2). We found strong correlations between both rBC mass concentration and rBC diameter with snow salinity. Therefore, we formulated the hypothesis of a salt-induced matrix effect interfering with the SP2 analysis. By replicating realistic salinity conditions, laboratory experiments indicated a dramatic six-fold reduction in observed rBC concentration with increasing salinity. In the salinity conditions tested in the present work (salt concentration below 0.4 g l$^{-1}$) the impact of salt on nebulization of water droplets might be negligible. However, the SP2 mass detection efficiency systematically decreased with salinity, with the smaller rBC particles being preferentially undetected. The high concentration of suspended salt particles and the formation of thick salt coating on rBC cores might have caused problems to the SP2 analog-to-digital conversion of the signal and incandescence quenching, respectively. Changes to signal acquisition parameters and laser power of the SP2 improved the mass detection efficiency, which, nonetheless, never attained unity values. The present work provides the evidence that high concentration of sea salt undermines the quantification of rBC in snow performed with the SP2. This interference was never reported and might affect future





analysis of rBC particles in snow collected, especially, over sea ice or coastal regions strongly affected by sea salt deposition.


## 1 Introduction

Black carbon aerosol (BC), produced by incomplete combustion of biomass and fossil fuels, is transported from mid-latitude source regions to the Arctic atmosphere (Schacht et al., 2019), where it influences the regional climate (Quinn et al., 2015). Once removed from the atmosphere, BC particles continue to affect

the Arctic radiative budget by directly decreasing the snow albedo (Dou and Xiao, 2016) and promoting snow grain growth (Skiles and Painter, 2017). In turn, the acceleration of the melting rate leads to earlier exposure of underlying surface. The overall process is usually called "snow albedo feedback", and might be considered among the strongest forcing mechanisms in the Arctic region (Hansen and Nazarenko, 2004; Flanner et al., 2007; Skiles et al., 2018).

Considering the climatic repercussions caused by BC in snow, the scientific community has been measuring the content of BC in snow across the Arctic for almost four decades (Clarke and Noone, 1985; Doherty et al., 2010; Dou and Xiao, 2016; Tørseth, 2019). Unfortunately, a standardized and universally accepted analytical technique does not yet exist. Generally, the wide variety of analytical approaches to measure BC in snow can be divided in offline and online methods. Considering the offline approach, BC

mass can be measured after melting and filtration of the snow sample via thermal-optical analysis (Hagler et al., 2007) or transmittance spectroscopy (Doherty et al., 2010). Alternatively, BC mass might be quantified, after the nebulization of the melted snow samples, with online techniques such as the photoacoustic technique (Schnaiter et al., 2019) or laser induced incandescence technique (Schwarz et al., 2012).

In the recent years the laser induced incandescence technique, more specifically the single particle soot photometer (SP2; Droplet Measurement Technologies, Longmont, CO, USA), was often deployed to quantify refractory black carbon particles (rBC; Petzold et al., 2013) in snow in various regions of the Arctic (Khan et al., 2017; Macdonald et al., 2017; Jacobi et al., 2019; Mori et al., 2019; Zhang et al., 2020). The rBC analytical procedure now generally includes three steps: 1) melting of the snow sample,

2) nebulization with pneumatic concentric nebulizer equipped with warming-cooling desolvating system (i.e. Marin-5 produced by Teledyne Technologies, Omaha, USA and APEX-Q produced by Elemental Scientific Inc., Omaha, USA), 3) sampling with the SP2. During nebulization, the melted sample is usually transported to the nebulizer at a constant flow rate via a peristaltic pump. The liquid is then broken into small droplets and suspended in a nebulization chamber by means of a pneumatic concentric

nebulizer. Once suspended, the solvent in the droplets is evaporated and removed with a warming cooling cycle. Several studies addressed the issue of reducing the losses of rBC during the nebulization phases by controlling liquid flow rate, gas flow rate and pressure and temperature cycle (Lim et al., 2014; Wendl et al., 2014; Mori et al., 2016; Katich et al., 2017). Overall, up to 75% of rBC mass is suspended from the sample, transported though the nebulizer and finally detected by the SP2 without the addition of

surfactants (Lim et al., 2014; Mori et al., 2016). Due to reduction of water density and viscosity, the doping with isopropyl alcohol increases the rBC mass nebulization efficiency to values close to unity





(Katich et al., 2017). It must be considered that, despite corrections for the nebulization efficiency, the degree of comparability with more traditional techniques (e.g. thermal-optical method) is still variable (Lim et al., 2014).

In the Arctic region, many snow samples were collected in coastal regions and over sea ice (Tørseth, 2019) where the sea salt components often dominate the snow chemical composition, especially in summer in presence of open waters (Krnavek et al., 2012; Jacobi et al., 2019). This is particularly relevant over sea ice, where sea salt aerosol, suspended as sea spray, can be deposited at the snow surface, while capillary upward migration of sea salt from the sea ice can lead to high salt concentration at the bottom

of the snowpack (Domine et al., 2004). It turned out that our surface snow samples, collected over the sea ice covered Fram Strait in summer 2017 within the PASCAL experiment, were highly affected by salt deposition, showing a wide range of salinity. The presence of salt might influence the nebulization of the sample, the analyte and solvent transport and even the analytical signal of certain analytical techniques such as inductively coupled plasma atomic emission spectroscopy (e.g. L. Sharp, 1988; Luis Todolí et al.,

2002; Burgener and Makonnen, 2020). This effect is commonly called "matrix effect". At the present time, the potential interference of sea salt during the analysis of rBC particles with the SP2 is not yet assessed.

Considering the high salinity of the snow samples, collected in the Fram Strait, a series of laboratory experiments were conducted to quantify the impact of sea salt on nebulization and rBC detection with the

SP2 instrument. This work aims to identify the importance of the salt matrix effect especially in the perspective of the MOSAiC (Multidisciplinary drifting Observatory for the Study of Arctic Climate, https://mosaic-expedition.org/) project, where hundreds of snow samples were collected, during the one-year long drift over the sea ice, for the analysis of rBC in snow with the SP2.

## 2 Technique

### 100 2.1 Snow sampling during the PASCAL expedition

The PASCAL expedition (Physical Feedbacks of Arctic Boundary Layer, Sea Ice, Cloud and Aerosol; Flores and Macke, 2018), organized within the framework of the AC3 project (Arctic Amplification: Climate Relevant Atmospheric and Surface Processes, and Feedback Mechanisms; Wendisch et al., 2018), was a shipborne field campaign on board of the *RV Polarstern*, the research icebreaker of the

Alfred Wegener Institute (Bremerhaven, Germany). The 25 surface snow samples discussed here were collected on the sea ice during the drift phase, which occurred between 3 June and 16 June 2017 in the Fram Strait between 10.0°E - 11.5°E and 81.7°N - 82°N at a distance from open leads between 0.5 km and 1 km. These samples were collected in the first 5 cm of the snow pack and stored frozen at -20°C in polypropylene tubes (Fisher Scientific GmbH, Schwerte, Germany) of 50 ml volume (typically 20-30 g

water content) until analysis. For each snow sample physical properties of the corresponding snow layer were also measured. The specific surface area (SSA) was measured with the IceCube instrument (A2 Photonic Sensors, Grenoble, France), the snow density with a custom-made density cutter and the snow temperature with a negative temperature coefficient one channel thermometer (model 101, Testo Ltd, Alton Hampshire, United Kingdom).





## 2.2 Instrumental setup during laboratory analysis

The deployed experimental setup for snow sample analysis is schematized in Figure 1. First, the snow samples were melted in a thermostatic bath at 25°C temperature. Immediately after melting, the electrical conductivity (κ) of the liquid sample was measured with a portable conductivity meter (model: Cond 3110, WTW, Xylem Analytics, Weilheim, Germany) equipped with a 2-electrode conductivity cell (LR 325/01). The probe was immersed in the sample and was rinsed with milliQ water before and after each measurement. Since κ increases with the concentration of ions in a solution
The sample was then pumped by a peristaltic pump through a liquid flow meter (model: Liquid Flow Meter SLI, Sensirion AG, Staefa, Switzerland) towards the Marin-5 nebulizer (Teledyne Technologies, Omaha, USA). In the Marin-5, the liquid sample is aerosolized by a concentric pneumatic nebulizer, such that the produced droplets are desolvated by a heating-cooling cycle (110°C-5°C). Finally, the dry aerosol (relative humidity below 30%; Katich et al., 2017) was pumped to the aerosol measuring instruments. More detailed description of the Marine-5 nebulizer can be found in Mori et al. (2016). The liquid (70 µl min$^{-1}$) and air (1 l min$^{-1}$) flow rates were selected to maximize the suspension of rBC mass following Katich et al. (2017) and kept constant during all experiments. No surfactants were added to the snow samples nor to the test suspensions/solutions. In this work, the aerosolized particles were directed to a Single Particle Soot Photometer (SP2; DMT, Longmont, USA), a Scanning Mobility Particle Sizer (SMPS; TSI; Shoreview, USA) and to the single-particle mass spectrometer ALABAMA (Aircraft-based Laser Ablation Aerosol Mass Spectrometer; Brands et al., 2011; Köllner et al., 2017; Clemen et al., 2020). Our SP2 sampled the aerosol directly from the Marin-5 exhaust and provided number concentration ($N_{rBC}$), mass concentration ($M_{rBC}$) and size distribution of rBC in the mass equivalent diameter ($D_{rBC}$) range of 70-1000 nm. The operation principle of the SP2 for atmospheric applications are given by Stephens et al. (2003), while a complete assessment on the performance of the SP2 during snow sample analysis can be found in Lim et al., (2014), Mori et al. (2016) and Katich et al. (2017). The incandescence detectors of the SP2 were calibrated with mass-selected fullerene soot (FS; Alfa Aesar, LOT: W08A039) as described in Laborde et al. (2012b). The SMPS measured the number concentration ($N$) and size distribution between 14 nm and 680 nm mobility diameter ($D_P$). Additionally, the ALABAMA was used to get additional information on the chemical composition of single particles with diameters between approximately 110 and 5000 nm (Clemen et al., 2020). Different chemical species were identified using characteristic marker ions of the mass spectra. Both the SMPS and ALABAMA sampling line included a factor 10 dilution. The transport losses of aerosol particles in the 15-1000 nm particle diameter range were estimated for the SP2 sampling line (0.30 m from the MARIN-5 exhaust), and for the ALABAMA-SMPS sampling line (1 m from the MARIN-5 exhaust). In general, the transport losses for both lines were negligible (<3%) for particles in the 30-1000 nm diameter range, while slightly highe
r losses (below 7%) were calculated for particles smaller than 30 nm. Considering the similarity of loss between the two sampling lines (SP2 and ALABAMA-SMPS) the transport losses are not considered in the rest of the present work.

## 2.3 Surface snow properties in the Fram Strait

### 2.3.1 Snow physical properties





Warm conditions were encountered during the drift, with the air temperature increasing from
approximately -4°C at the beginning of the campaign to approximately -1°C (with one-minute average
values up to 2°C) at the end of the drift. As a consequence
of temperature increase, the specific surface area (SSA) decreased from 70 $m^2$ $kg^{-1}$ to 5 $m^2$ $kg^{-1}$ and snow
density increased from 280 kg $m^{-3}$ to 350 kg $m^{-3}$. The electrical conductivity of surface snow decreased
from values above 1000 μS $cm^{-1}$ to κ values below 10 μS $cm^{-1}$ towards the end of the campaign .The
decrease of SSA and increase of snow density during the drift phase of the PASCAL expedition indicates
the occurrence of melt-refreeze cycles (Haas et al., 2001; Massom et al., 2001; Domine et al., 2007), and
the downward migration of soluble salts with percolating water explains the decrease of κ (Domine et al.,
2004; Doherty et al., 2013). Considering the wide variability of electrical conductivity, the dataset was
organized in 5 salinity classes (Sn). The samples with the lowest conductivity (κ ≈ 5-10 μS $cm^{-1}$)
accounted for 38% of total samples and were mainly collected after the melting onset and grouped into
S1. S2 includes the samples collected at the snow melt which showed κ values between approximately
20-30 μS $cm^{-1}$ and accounted for 17% of the samples. The samples characterized by a κ value above 200
μS $cm^{-1}$ and below 2000 μS $cm^{-1}$ represented the 46% of the total number of the collected probes and
were organized into three different salinity classes (S3, S4 and S5). The most saline snow sample (κ =
3600 μS $cm^{-1}$) was excluded from further analysis. The mean and boundaries of κ values defining the five
salinity classes are listed Table 1.

### 2.3.2 Relationship between salinity of snow samples and particle aerosolization

In this part of the work we investigate the potential relationship between electrical conductivity, which is
used here as a proxy for salinity, and the properties of aerosolized particles. Note that both $N$ and $N_{rBC}$
presented in this section are not corrected for the nebulization efficiency of the Marin-5 nebulizer.
Under fixed nebulization conditions (constant liquid sample flow and gas flow), a large number of
particles (droplet residues) were suspended by the nebulization process. $N$ increased with the electrical
conductivity of the sample from 6.1*$10^5$ $cm^{-3}$ in low conductivity samples (κ < 10 μS $cm^{-1}$) to 1.5*$10^6$ $cm^{-3}$
for samples showing conductivity values above 1000 μs $cm^{-1}$ (Figure 2, Table 1). Similar trend was
recently observed by Rösch and Cziczo (2020). The correlation of $N$ with κ suggest that most of the
aerosolized particles might be composed by sea salt, which is expected to be the major solute in the sea
water. As indicated in Table 1, the size of the aerosolized particles (geometric mean of the number size
distribution; $GD_P$) increased with κ from 27 nm (S1, κ < 10 μS $cm^{-1}$) to 89 nm (S5, κ>1000 μS $cm^{-1}$).
The $GD_P$ – κ relationship and absence of multiple modes in the aerosol size distribution (Figure S1)
supports the assumption of sea salt controlling the aerosol composition. The increase of particle diameter
with liquid concentration of soluble inorganic salts was already observed (Clifford et al., 1993) and found
to be mainly caused by the higher concentration of salt in the primary aerosolized droplets. Next to it,
particle growth caused by coalescence and promoted by the high number concentration of aerosolized
particles might also contribute to the diameter shift (Olivares and Houk, 1986). The shift of the size
distribution mode into the size detection range of the SMPS might also contribute to the $N$-κ positive
correlation shown in Figure 2.
The ALABAMA measurements confirmed the predominant presence of sea water components such as
sodium chloride (NaCl) and magnesium (Mg) over other chemical species shown in Figure 3. The particle



fraction (*PF*) of NaCl-containing particles increased from roughly 30 % for the lowest salinity class to 60-80 % of all analyzed particles for S2, S3, S4 and S5. Similar increase was observed for Mg-containing particles. Other particle species, e.g. non-sea-salt (nss) nitrate and sulfate, were only abundant in samples with low conductivity. For salinity class S4, the fraction of NaCl- and Mg-containing particles is significantly lower compared to other classes with $\kappa$>10 µS cm$^{-1}$ which is due to one sample of S4

containing more of the other particle species.

SP2 measurements indicated a monotonic decrease of $N_{rBC}$ with conductivity (Figure 2, Table 1), opposite to $N$. Considering the salinity classes, the rBC number concentration decreased from approximately 70 cm$^{-3}$ in S1 to 1.7 cm$^{-3}$ in S5. Additionally, the number size distribution of detected rBC particles showed a shift to larger diameter as function of salinity (Figure S1). The resulting rBC geometric mean diameter

($GD_{rBC}$), calculated from the number size distribution, increased from approximately 90 to 120 nm from S1 to S5 (Table 1). One major aspect surge from these first results: rBC-containing particles represent the small minority of the aerosol population nebulized from the snow samples. In fact, the number fraction of rBC particles ($F_{rBC}$) decreased with salinity from 1.1*10$^{-2}$ % in S1 to 1.1*10$^{-4}$ % in S5 (Table 1). Moreover, elemental carbon (EC-) containing particles were found in less than 1% of all analyzed

particles by ALABAMA (Figure 3). Hence, considering the remarkable concentration of total particles and the minor fraction of rBC particles, bench experiments were designed to reproduce the salinity conditions of PASCAL snow samples in order to investigate the potential interference mechanisms of salt on rBC detection by the SP2.

### 2.4   Reproducing realistic snow samples conditions in laboratory experiments

The laboratory experiments aimed to reproduce BC snow concentration representative of generic Arctic conditions, and the specific salinity conditions representative of PASCAL snow samples. It is important to note that the salt concentrations explored in the present work do not represent realistic conditions encountered in continental or mountain regions were sea salt aerosol deposition is not dominant. The ability of reproducing such conditions and practical limitations are presented as following.


#### 2.4.1   Sample preparation

Fullerene soot (FS; Alfa Aesar, LOT: W08A039) was used as proxy for ambient black carbon. FS is a well characterized standard for SP2 calibration (Gysel et al., 2011; Laborde et al., 2012a) and is consensually accepted as reference standard for ambient black carbon (Moteki and Kondo, 2010;

Baumgardner et al., 2012). Three different inorganic salts were initially chosen to replicate the conductivity array of the snow samples: sodium chloride (NaCl, Fluka), potassium chloride (KCl, Fluka) and ammonium sulfate ((NH$_4$)$_2$SO$_4$, Sigma Aldrich).

The proportionality between conductivity and mass concentration of the salts was first assessed (Figure S2). All the saline solutions showed a linear relationship between conductivity and mass concentration,

with high correlation coefficient (r$^2$>0.98). We have chosen the concentration range of the dissolved salt to be between approximately 3 mg l$^{-1}$ and 870 mg l$^{-1}$ to match the salinity values between the lowest and the highest boundaries of the S1-S5 classes. As in the case of snow samples, the number concentration of aerosolized particles (measured by the SMPS in the size range of 14-680 nm) increased with salinity (Figure 4) from approximately 6*10$^5$ cm$^{-3}$ at S1 ($\kappa \approx$ 7 µS cm$^{-1}$) to approximately 1.7*10$^6$ cm$^{-3}$ at S5 ($\kappa \approx$



1450 µS cm⁻¹). The number concentration of particles produced by nebulization of saline solutions is similar to the snow samples with difference below 10% for the S1-S4 classes and below 15% for S5. The increase of particle diameter with conductivity observed in the snow samples was closely replicated by the saline solutions (Figure 5). The increase of concentration and diameter of total suspended particles with salinity observed in the snow samples was successfully reproduced with salt standards, indicating
that the use of inorganic salts to simulate the snow saline conditions was justified. Due to the similarity of results between the different salts, and to the fact that NaCl was the major component of PASCAL samples (Figure 3), NaCl was chosen as the reference salt for all the experiments presented in the rest of the study.

### 2.4.2  Salt induced limitations to SP2 operation

Although ambient salinity conditions were successfully reproduced, salt presence affected the operation of the SP2. Hence, the BC and salt concentration range was limited, compared to ambient conditions, in order to ensure a correct SP2 analysis.

#### 2.4.2.1  SP2 laser beam shadowing

During the snow-sample analysis, the large light extinction of the dense aerosol produced from one extremely saline sample ($\kappa = 3600$ µS cm⁻¹) caused a marked but temporary decrease of the YAG power (approximately 50% reduction). The laser power completely and immediately recovered after switching to clean air sampling.  Despite this extreme event, in the salinity range of our snow samples ($\kappa < 1652$ µS cm⁻¹), the YAG power showed only a 2% decrease with increasing salinity. This effect on net laser
intensity in the cavity might, thus, be important for extremely saline solutions, but negligible for solution or samples with $\kappa$ below 1500 µS cm⁻¹ at the rates we nebulized the liquid.

#### 2.4.2.2  SP2 flow system

The prolonged sampling of saline solution also affected the flow system of the SP2. First, the continuous injection of high concentration salt particles caused the obstruction of the aerosol jet tubing directing the sampled air into the SP2 detection chamber. The solid salt obstruction could be removed by partial and rapid immersion in water of the jet assembly followed by drying with compressed air. However, this procedure caused a long disruption of the analysis sequence and can alter the jet-laser alignment. The first
set of experiments were based on FS suspensions at three mass concentrations: 1 µg l⁻¹, 5 µg l⁻¹ and 10 µg l⁻¹. The clogging frequency was higher at low FS concentration, when the sampling time (needed to acquire minimum of 30 000 recorded particles) was substantially longer than for more FS concentrated suspensions (Figure S3). Although experiments with suspensions at higher FS concentration were less frequently affected by clogging issues the prolonged nebulization (longer than 30 minutes) of saline
solutions with electrical conductivity equal and above 800  µs cm⁻¹ caused systematic clogging of the SP2 aerosol jet tubing. Hereafter, only the results obtained from the nebulization of FS suspension at 10 µg l⁻¹ will be presented and discussed. This value is fairly representative of central Arctic Ocean snow BC concentration range (3-15 µg g⁻¹) measured with transmittance spectroscopy (Doherty et al., 2010). In order to reduce the clogging occurrence, the upper limit of FS suspension's $\kappa$ was always set to 800 µS
cm⁻¹ (approximately 0.4 g l⁻¹ of NaCl). Considering our inability of reproducing the most extreme sample


conditions (low BC and high salt concentration), the SP2 results acquired during the analysis of S5 samples are affected by unknown bias.

One also has to pay thorough attention to the particle drying during prolonged saline solution measurements. The transport of the remaining moisture through the SP2 might not directly impact the detection of rBC particles, but it might damage the SP2 pump and inject moist sheath and purge air in the SP2, potentially contaminating the SP2 optics. To avoid this problem, we have installed an additional dryer at the SP2 exhaust and a water trap immediately after the SP2 pump in addition to the standard dryer usually installed between the pump and the SP2 purge-sheath flow.

## 2.5 Salt impact on the rBC mass quantification by the SP2

In this section we investigate the possible impact of salt on the SP2 rBC quantification using the laboratory generated snow sample proxies. When nebulizing liquid samples, the overall mass quantification efficiency of rBC ($\varepsilon$) can be calculated as the mass of rBC pumped into the nebulizer per unit time divided by the mass of rBC coming out of the nebulizer in aerosol form and detected by the SP2 per unit time (Katich et al., 2017):

$$\varepsilon = \frac{M_{rBC-SP2}}{M_{rBC-smp}} * \frac{F_{Gas}}{F_{Liq}} \tag{1}$$

Where $M_{rBC-SP2}$ and $M_{rBC-smp}$ is the mass concentration of rBC particles quantified by the SP2 and the mass concentration of rBC particles in the liquid sample, respectively. $F_{Gas}$ is the gas flow rate applied to nebulizer and $F_{Liq}$ is the liquid flow rate of sample pumped into the nebulizer. The $\varepsilon$ values obtained for various soot standards suspended in pure water and nebulized with the Marin-5 nebulizer vary between 0.5 and 0.6 (Mori et al., 2016; Katich et al., 2017), while $\varepsilon$ of 0.75 was calculated by Lim et al. (2014) using a APEX-Q nebulizer. The $\varepsilon$ calculated in the present work for non-saline FS suspension varied 0.58 and 0.66. These values are in good agreement with the previous studies, indicating a good reproducibility of nebulization conditions during the laboratory experiments. However, the $\varepsilon$ monotonic decrease of $\varepsilon$ down to values between 0.33 and 0.21 for the most saline FS suspensions ($\kappa = 800$ cm$^{-1}$) clearly indicated a salt induced bias on the mass quantification efficiency. $\varepsilon$ may be decomposed in three distinct contributions: the mass nebulization efficiency ($\varepsilon_{Neb}$), the transport efficiency ($\varepsilon_{Trn}$) and the mass detection efficiency of the SP2 ($\varepsilon_{SP2}$). $\varepsilon_{Neb}$ describes the mass of rBC pumped into the nebulizer per unit time divided by the mass of rBC coming out of the nebulizer in aerosol form per unit time. Hence, $\varepsilon_{Neb}$ depends on the suspension efficiency of the liquid sample in liquid droplets and on the transport efficiency of the wet and dry aerosol within the nebulizer. $\varepsilon_{Trn}$ describes the mass of rBC pumped out the nebulizer per unit time divided by the mass of rBC reaching the SP2 sampling inlet per unit time. $\varepsilon_{Trn}$ is controlled by diffusion and sedimentation losses and depends on flow rate and particle size but also on length and bends of the sampling line. $\varepsilon_{SP2}$ describes the mass of rBC in aerosol form being introduced to the SP2 laser per unit time divided by the mass of rBC in aerosol form detected by the SP2 per unit of time. Considering typical atmospheric SP2 operation, $\varepsilon_{SP2}$ was measured to be of 100% in the SP2 size detection range, being independent from the presence of non-absorbing and non-refractory atmospheric components (Schwarz



et al., 2010). It must be considered that past studies always assumed a 100% $\varepsilon_{SP2}$ and $\varepsilon_{Trn}$. Therefore, $\varepsilon$ (calculated exactly as in this study according to Equation (1) was usually addressed as nebulization efficiency. The decrease of $\varepsilon$ with salinity indicates that, in the presence of salt, this assumption might not be realistic. The potential impact of salt presence on the nebulization and transport efficiency, and on the SP2 detection efficiency will be addressed in the following subsections.

### 2.5.1   Nebulization and transport efficiency

Inorganic salt can alter water properties such as viscosity and surface tension, modifying the size distribution of droplets produced in the spray chamber and affecting the nebulization efficiency of various analytes (Todolí et al., 2002). High surface tension and viscosity cause an increase of the mean diameter of the liquid droplets suspended by pneumatic nebulizers, decreasing the transport efficiency (L. Sharp, 1988). NaCl concentration above 1 g l$^{-1}$ causes a significant increase of droplet diameter (Clifford et al., 1993) and a decrease of transport efficiency (Sötebier et al., 2016). However, previous studies on pneumatic nebulizers (Dubuisson et al., 1998; Cano et al., 2002) indicated that salt matrix effect on the liquid droplet diameter and transport at concentration below 1 g l$^{-1}$ is negligible. It must be noted that, as summarized by Todolí et al. (2002), several studies have reported different or contradictory results on the impact of matrix effect during nebulization process with pneumatic nebulizers. Thus, the understanding of processes causing diameter increase and efficiency decrease remains uncertain. The viscosity and surface tension of our NaCl solutions was extrapolated from Lide (1995) and Ozdemir et al. (2009), respectively. 0.8 g l$^{-1}$ of NaCl, representative of S5 samples ($\kappa = 1400$ µS cm$^{-1}$), causes a small increase of both viscosity (+0.08%) and surface tension (+0.03%) compared to pure water. Our calculations and the results of Dubuisson et al. (1998) and Cano et al. (2002) indicate a negligible change of water properties and thus nebulization efficiency for the salinity levels tested in the laboratory experiments ($\kappa$ below 800 µS cm$^{-1}$, NaCl concentration below 0.4 g l$^{-1}$). Considering the dominant presence of NaCl in the snow samples (Figure 3), our calculations should reflect snow samples conditions. In order to support our calculations, the mass quantification efficiency was calculated for NaCl solutions ($\varepsilon_{NaCl}$) at different salt concentration from SMPS measurements (14 nm $< D_p <$ 680 nm). The aerosolized mass concentration of the NaCl was calculated assuming spherical particles with a density of 2.17 g cm$^{-3}$. Thanks to the factor 10 dilution, which granted SMPS operation below the highest limit of the detection and low RH (<30%), we assumed a SMPS detection efficiency of 100% in the 14-680 nm diameter range. Thus, $\varepsilon_{NaCl}$ should solely be affected by nebulization efficiency. $\varepsilon_{NaCl}$ remained very stable (range of 0.57-0.58) from low ($\kappa = 25$ µS cm$^{-1}$) to high ($\kappa = 800$ µS cm$^{-1}$) salinity. In turn, this suggest that the nebulization efficiency of rBC particles is independent from salinity (salt concentration below 4 g l$^{-1}$ or electrical conductivity below 800 µS cm$^{-1}$. Regarding transport efficiency, diffusion and sedimentation losses during transport, from the nebulizer exhaust to the SP2 (0.3 m of length), were estimated for particles in the SP2 detection range (70-1000 nm) to remains well below 2%; hence not taken into account in forthcoming calculations.

### 2.5.2   SP2 detection efficiency

In this section we will investigate the consequences of high number particle density transiting the laser beam on the SP2 data acquisition system as function of different acquisition settings. On second instance, the potential quenching of incandescence caused by the formation of thick salt coating on rBC cores will





be addressed. $\varepsilon_{SP2}$ is calculated as the ratio between $M_{rBC}$ of the NaCl-doped suspensions at a certain
conductivity ($\kappa > 0$ μS cm$^{-1}$; $\kappa_x$) and the $M_{rBC}$ of the non-saline reference suspension ($\kappa \approx 0$ μS cm$^{-1}$; $\kappa_0$)
as:

$$\varepsilon_{SP2}(\kappa_x) = \frac{M_{rBC}(\kappa_x)}{M_{rBC}(\kappa_0)}$$

(2)

A size dependent $\varepsilon_{SP2}$ is defined as the ratio of the mass size distribution at a specific conductivity ($\kappa_x$)
over the mass size distribution of non-saline suspensions ($\kappa_0$) at a certain rBC diameter ($D_{rBC}$):

$$\varepsilon_{SP2}(\kappa_x, D_{rBC}) = \frac{dM_{rBC}(\kappa_x, D_{rBC})}{dM_{rBC}(\kappa_0, D_{rBC})}$$

(3)

where d$M_{rBC}$ represents the mass concentration of rBC particles contained in each individual diameter bin
of the rBC mass size distribution.

### 2.5.2.1 Sensitivity of data acquisition to sample salinity

The frequency of detected scattering events increased up to 17500 counts per second for saline solutions
with 250 μS cm$^{-1}$ electrical conductivity (Figure S4). Considering the high frequency of concomitant
events (multiple particles passing through the laser beam during one acquisition period), this number is
most likely underestimated. Operating the SP2 at these overloaded condition might lead to data loss
caused by the limitation of the analogue to digital converter and computational power of the instrument's
PC. This is possibly responsible for the observed exponential decrease of the frequency of detected
scattering events for saline solutions with $\kappa$ values above 250 μS cm$^{-1}$. Concomitant particles and trigger
hysteresis (used to reduce some types of data artifacts in the SP2) issues do not only reduce the acquisition
of scattering events, but also cause the decrease of detected incandescence events' frequency (Figure S4).
Considering the low concentration of rBC particles and high concentration of non-rBC particles, the
acquisition settings become extremely important to avoid non-detection of individual incandescence
signals. The importance of triggering choices was, hence, tested with 10 μg l$^{-1}$ FS concentration and
increasing NaCl concentration (conductivity range of 0 - 800 μS cm$^{-1}$). The mass detection efficiency was
then measured with either the signal acquisition triggering off the high-gain scattering channel ($\varepsilon_{SP2-Tsc}$)
or off the broad-band high-gain incandescence channel ($\varepsilon_{SP2-Tin}$).

First, the result obtained by triggering on the scattering detector (the typical setup when operating the SP2
for atmospheric observations) will be discussed. $\varepsilon_{SP2-Tsc}$ steadily decreased with conductivity to 0.34 at
800 μS cm$^{-1}$, while the rBC mean diameter increased from 180 at 0 μS cm$^{-1}$ nm to 207 nm at 800 μS cm$^{-1}$ (Figure 6) The increasing rBC mean diameter suggests the size dependency of $\varepsilon_{SP2-Tsc}$ (see rBC size
distribution in Figure S5). $\varepsilon_{SP2-Tsc}$ systematically increased with $D_{rBC}$ across the entire salinity range tested
in the present study (Figure 7a). At lower conductivity ($\kappa < 200$ μS cm$^{-1}$), $\varepsilon_{SP2-Tsc}$ of particles smaller than
100 nm increased from approximately 0.4-0.6 to values above 0.7 for rBC particles larger than 200 nm.
The difference of $\varepsilon_{SP2-Tsc}$ across the rBC size distribution was remarkable at higher conductivity ($\kappa > 600$
μS cm$^{-1}$), where $\varepsilon_{SP2-Tsc}$ increased from 0.1-0.2 for $D_{rBC}$ below 100 nm to 0.4-0.6 for $D_{rBC}$ above 300 nm.
It is thus evident that not only the mass, but also the size distribution, of rBC in saline samples might be


strongly biased when the standard atmospheric configuration of the SP2 is implemented for snow measurements with significant salt content. Below 200 µS cm⁻¹, no significant difference could be identified between $\varepsilon_{\text{SP2-Tsc}}$ and $\varepsilon_{\text{SP2-Tin}}$. This result indicates that triggering settings are not crucial for relatively clean snow samples. Nevertheless, incandescence triggering allowed recovering 20-50% more rBC mass compared to scattering triggering at higher salinity ($\kappa$>200 µS cm⁻¹). Similar to $\varepsilon_{\text{SP2-Tsc}}$, $\varepsilon_{\text{SP2-Tin}}$ showed an increasing trend with $D_{\text{rBC}}$ at all considered electrical conductivities (Figure 7b). We have to

note here that for $D_{\text{rBC}}$ larger than 400 nm the number of the detected rBC particles in the last bins of the size distribution was extremely low, therefore the statistical uncertainty caused unrealistic $\varepsilon$SP2 values above 1. Considering the results presented here, triggering setting does not appear to substantially modify the size dependency of the SP2 detection efficiency. Nevertheless, scattering trigger should be avoided in future studies when analyzing rBC in snow samples. Diluting these saline samples to achieve only single

particles per detection in the SP2 (Katich et al., 2017), would likely reduce detection problem only for samples with $\kappa$ above 200 µS cm⁻¹ (NaCl concentration above 0.1 g l⁻¹).

### 2.5.2.2 Incandescence quenching

In presence of very thick coatings encapsulating rBC cores, the laser beam of the SP2 might not have enough power to penetrate the coating, warm the rBC core, evaporate the coating and finally trigger incandescence. This phenomenon will be called incandescence quenching. During nebulization of saline samples, the salt, contained in each droplet, will remain on the rBC particles with water evaporation and lead to the formation of rBC cores encapsulated by thick coatings.

As rough estimation, we calculated the theoretical coating thickness of spherical rBC particles having a diameter of 100 nm, 200 nm, 300 nm, 400 nm, 500 nm contained in 8 µm droplets as function of salinity. We assumed the presence of a single rBC particle per droplet, a concentric core-shell geometry and NaCl density of 2.16 g cm⁻³. A diameter of 8 µm represents the peak of the primary droplet number size distribution suspended by various concentric pneumatic nebulizers working with a gas flow rate of 1 L

min⁻¹ and a liquid flow range of 11-20 µL min⁻¹ (Burgener and Makonnen, 2020). The calculated coating thickness assumptions can be found in Table S2. The thickest coatings are expected for the smallest rBC cores (100 nm of diameter) with values increasing from 59 nm at 50 µS cm⁻¹ to 197 nm at 800µS cm⁻¹. Coating thickness for the largest cores considered here (500 nm) were significantly smaller, having values below 55 nm for most of the considered NaCl concentration. This calculation shows that incandescence

quenching would have a stronger influence on the detection of smaller rBC particles. This phenomenon could explain the measured $\varepsilon_{\text{SP2}}$ values' diameter dependence shown in Figure 7. The calculated coating thickness is only a rough assumption, since its value strongly depends on the size of the nebulized droplets. Assuming 12 µm and 5 µm droplet diameter (upper and lower mode of the size distribution shown in Burgener and Makonnen, 2020) will increase and decrease the coating thickness value by

approximately a factor 2, respectively (Table S1). Laboratory experiments showed that relatively thin coatings (coating-rBC mass ratio below 3.5) do not have any significant negative impact on the SP2 detection efficiency (Schwarz et al., 2010). According to the coating calculation presented above (droplet diameter 8 µm), coating-rBC mass ratio above 3.5 are expected for rBC particles with diameter below 300 nm. For 100 nm particles, coating-rBC mass ratio were calculated to exceed a value of 7 already at





50 µS cm$^{-1}$. Considering our calculation, and results obtained by Schwarz et al. (2010), incandescence quenching might be particularly important for the smallest rBC particles.

To further investigate the potential quenching effect of salt, FS suspension (FS concentration of 10 µg l$^{-1}$) with increasing NaCl concentration was additionally nebulized and analyzed with the SP2 operating at maximum YAG-laser power and signal acquisition triggered on the incandescence signal. High YAG-
laser power is expected to vaporize thicker coatings and to increase the mass detection efficiency of rBC (Schwarz et al., 2010). During standard operational conditions and for the above presented results, the pump laser was operated with a current of 2800 mA. For the following experiments, the pump-laser current was increased to its upper limit (3200 mA), leading to an increase of the YAG-laser power output from 5.2 V to 6 V. The mass detection efficiency and its size distribution, here called $\varepsilon_{SP2-Ymax}$, were
calculated according to Equation 1 and Equation 2, respectively. $\varepsilon_{SP2-Ymax}$ decreased monotonically from 0.83 at 50 µS cm$^{-1}$ to 0.55 at 800 µS cm$^{-1}$ (Figure 6). Although the negative correlation between $\varepsilon_{SP2}$ and $\kappa$ was still present, $\varepsilon_{SP2-Ymax}$ was systematically (10% on average) higher than $\varepsilon_{SP2-Tin}$ across the full salinity range ($\kappa > 0$ µS cm$^{-1}$). If we look at the size dependent efficiency in Figure 7c, we see that the higher laser power did not improve the detection efficiency in the 70-100 nm $D_{rBC}$ range. An improvement
of $\varepsilon_{SP2-Ymax}$ were observed for rBC particles with $D_{rBC}$ larger than 100-150 nm compared $\varepsilon_{SP2-Tin}$. It is evident that operating the pump laser at the maximum performance we could achieve does not impact the overall decreasing trend of $\varepsilon_{SP2}$ with $\kappa$, and does not ensure homogeneous size detection of rBC particles. These results are consistent with quenching of incandescence due to coatings, with stronger reductions in smaller and more thickly-coated rBC cores. Diluting these samples to electrical conductivity values below
50 µS cm$^{-1}$ (salt concentration of 0.03 g l$^{-1}$), when $\varepsilon_{SP2}$ is higher than 80%, might significantly reduce incandescence quenching. However, dilution factor will linearly increase with conductivity up to a factor 16 for very saline samples ($\kappa=800$ µS cm$^{-1}$). Considering the low BC concentration generally observed in Arctic snow, dilution will strongly increase analysis time.

## 3  Conclusions

Laboratory experiments were conducted to assess the interference caused by inorganic salt with SP2 measurements applied to the quantification of rBC mass in saline snow samples. These experiments were designed to reproduce the salinity conditions of snow samples collected over the sea ice covered Fram Strait in summer 2017 during the PASCAL drift shipborne campaign. These salt concentrations might be exclusively encountered in snow collected in coastal areas or over sea ice in vicinity of open water. The
total mass quantification efficiency ($\varepsilon$), which consists of the MARIN-5 nebulization efficiency ($\varepsilon_{Neb}$) and the SP2 detection efficiency ($\varepsilon_{SP2}$), was strongly influenced by the salinity of fullerene suspension. Compared to $\varepsilon$ of 0.58-0.66 for non-saline fullerene soot suspension, high salinity (electrical conductivity of 800 µS cm$^{-1}$ and NaCl concentration of 0.4 g l$^{-1}$) caused $\varepsilon$ drop to 0.2-0.3 in laboratory testing. Considering that the slight increase of density and viscosity of water and the stable $\varepsilon$ of 0.57-0.58 for
NaCl solutions, $\varepsilon_{Neb}$ could be considered to be independent from salinity below NaCl concentration of 1 g l$^{-1}$. The sensitivity of $\varepsilon_{SP2}$ to different SP2 settings was also tested in order to verify: 1) the impact of high number concentration of non-rBC particles on the SP2 signal acquisition; 2) the incandescence quenching produced by thick coatings. Different SP2 settings were tested, but none of them allowed



recovering the entirety of rBC mass, with maximum $\varepsilon_{SP2}$ of approximately 0.85 and 0.55 for low- and high- saline samples; respectively. The SP2 detection efficiency was also found to strongly depend on the rBC particles' diameter. Overall, rBC particles below 100 nm might be substantially undetected, showing $\varepsilon_{SP2}$ below 0.2 for the most saline samples (electrical conductivity of 800 $\mu$S cm$^{-1}$ and NaCl concentration of 0.4 g l$^{-1}$) compared to values above 0.6 for rBC particles larger than 200 nm. As a consequence of the variable detection efficiency, the rBC mass of the PASCAL snow samples is affected by a high degree of

uncertainty, and underestimated significantly depending on snow salinity. By operating the SP2 in its optimal triggering setup and maximum YAG-laser power, the rBC mass in the less saline samples, collected after melting (electrical conductivity below 50 $\mu$S cm$^{-1}$) and representing 55% of total probes, are underestimated by a maximum of 17%. In the most saline samples collected before melting (electrical conductivity above 800 $\mu$S cm$^{-1}$ and NaCl concentration above 0.4 g l$^{-1}$) the rBC mass might be

underestimated at least by 45%. Moreover, it is important to note that the nebulization and sampling of saline samples might have negative impact on the SP2 performance. Extremely saline samples could cause a temporary drop of the YAG-laser output power, while prolonged sampling of even low salinity samples could cause the clogging of the SP2 aerosol-spray tubing.

The work demonstrates the influences of sea salt on the performances of the SP2 system during the

analysis of snow samples containing notable amount of inorganic salt. We aim to draw attention on a specific technical analytical issue that was never explicitly addressed beforehand, and that might alter the analysis of future studies conducted in marine regions. Additional work needs to be performed in order to minimize the impact of the matrix effect and to address the potential interference of salt on other BC measuring techniques.


**Special issue statement**. This article is part of the special issue "Arctic mixed-phase clouds as studied during the ACLOUD/PASCAL campaigns in the framework of (AC)3 (ACP/AMT/ESSD inter-journal SI)". It is not associated with a conference.

**Acknowledgements**: We gratefully acknowledge the funding by the Deutsche Forschungsgemeinschaft (DFG, German Research Foundation)– project ID 268020496 – TRR 172, within the Transregional Collaborative Research Center "ArctiC Amplification: Climate Relevant Atmospheric and SurfaCe Processes, and Feedback Mechanisms (AC)3". OE acknowledges funding by the Max Planck Graduate School (MPGC).




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





## Tables

**Table 1 Properties of snow samples grouped in conductivity classes. Conductivity of melted samples measured between 20 and 25 °C. Mass mixing ratio ($M_{rBC}$) and geometric mean of the number and mass sized distribution ($GD_{rBC}$) measured between 60 and 1000 nm of $D_{rBC}$. Number concentration of the total aerosolized particles ($N$) measured in the mobility diameter range 14-685 nm. Geometric mean of the total aerosol number size distribution ($GD_P$). Number fraction of rBC particles ($F_{rBC}$).**

| Salinity class | Fraction sample [%] | Conductivity [μS cm⁻¹] | | | N [cm⁻³] | | | $N_{rBC}$ [cm⁻³] | | $F_{rBC}$ [%] | $D_P$ [nm] | $D_{rBC}$ [nm] |
|---|---|---|---|---|---|---|---|---|---|---|---|---|
| | | Lower boundary | Upper boundary | Mean | Mean | SD | | Mean | SD | Mean | Geometric mean number | Geometric mean number |
| S1 | 38 | 5.3 | 10.3 | 7.07 | 6.1*10⁵ | 1.3*10⁵ | | 69.6 | 34.04 | 1.1*10⁻² | 27.3 | 87.6 |
| S2 | 17 | 19.6 | 33.1 | 24.9 | 9.5*10⁵ | 1.0*10⁵ | | 13.1 | 1.64 | 1.4*10⁻³ | 34.4 | 89.2 |
| S3 | 21 | 219 | 343 | 266 | 1.3*10⁶ | 6.2*10⁴ | | 6.81 | 2.03 | 5.2*10⁻⁴ | 57.6 | 94.2 |
| S4 | 13 | 466 | 533 | 497 | 1.4*10⁶ | 6.7*10⁴ | | 4.45 | 3.71 | 3.1*10⁻⁴ | 72.1 | 106 |
| S5 | 13 | 1275 | 1652 | 1424 | 1.5*10⁶ | 9.0*10⁴ | | 1.71 | 0.26 | 1.1*10⁻⁴ | 89.0 | 120 |

**Table 2 Summary of the analysis of fullerene soot suspensions at a concentration of 10 μg l⁻¹ doped with NaCl for different SP2 settings: Tsc = scattering trigger and standard YAG-laser power , Tsc = scattering trigger and standard YAG-laser power; Tin = incandescence trigger and standard YAG-laser power; Ymax = incandescence trigger and maximum YAG-laser power. SP2 detection efficiency ($ε_{SP2}$), geometric mean of the rBC mass size distribution ($GD_{rBC}$) listed as function of increasing electrical conductivity (κ) measured between 24 and 25°C.**

| κ [μS cm⁻¹] | $ε_{SP2}$ [-] | | | $GD_{rBC}$ [nm] | | |
|---|---|---|---|---|---|---|
| | Tsc | Tin | Ymax | Tsc | Tin | Ymax |
| 0 | 1 | 1 | 1 | 183 | 180 | 183 |
| 50 | 0.77 | 0.79 | 0.83 | 188 | 190 | 191 |
| 100 | 0.74 | 0.75 | 0.79 | 193 | 192 | 193 |
| 200 | 0.58 | 0.57 | 0.71 | 195 | 195 | 197 |
| 400 | 0.47 | 0.56 | 0.61 | 198 | 200 | 204 |
| 600 | 0.41 | 0.51 | 0.57 | 203 | 206 | 207 |
| 800 | 0.34 | 0.5 | 0.54 | 206 | 211 | 211 |






**Figures**

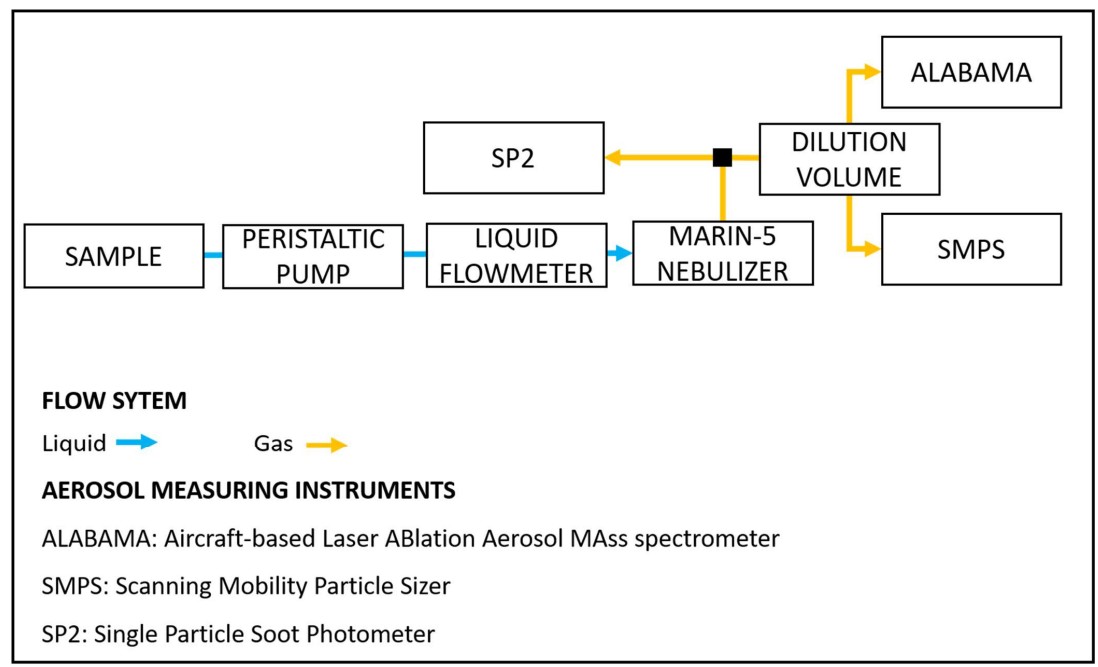

**Figure 1 Schematics of the instrumental setup deployed to analyze the PASCAL snow samples and to perform the laboratory test**
**experiments. ALABAMA not available for laboratory test experiments.**



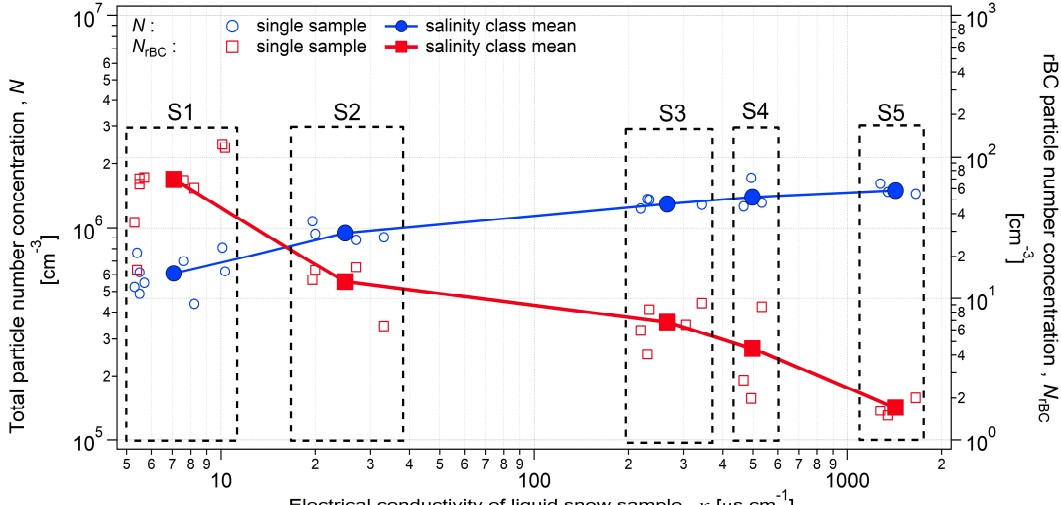

**Figure 2** Number concentration of the total particles ($N$) and rBC particles ($N_{rBC}$) produced from the nebulization of PASCAL snow samples as function of the electrical conductivity of melted snow samples. Boxes indicates the salinity classes (Sn). $N$ measured with the SPMS in the 14-680 nm diameter range nm. $N_{rBC}$ measure with the SP2 in the 70-1000 nm diameter range.



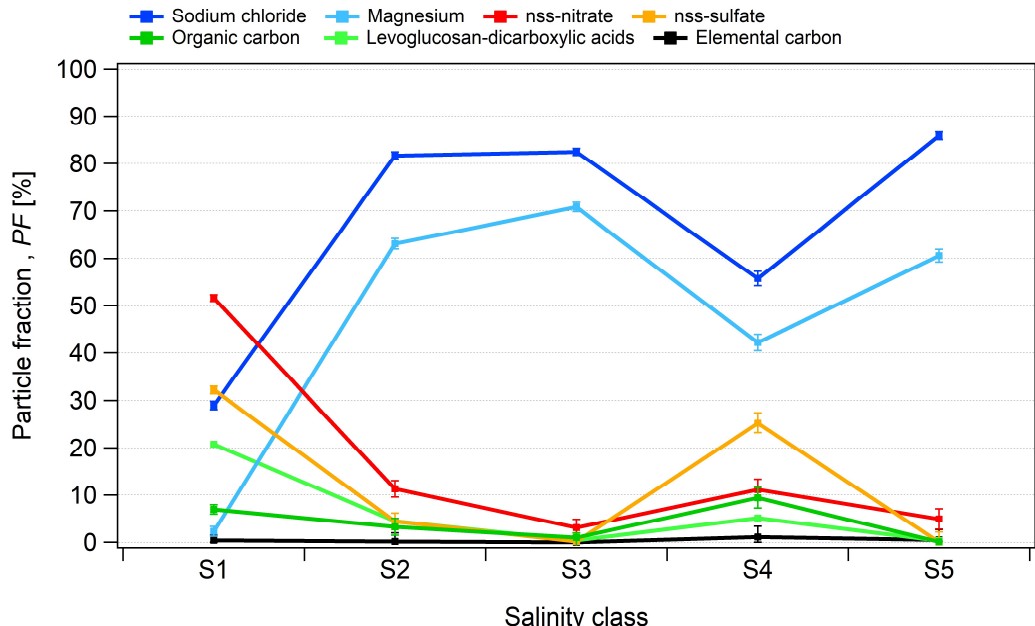

**Figure 3 Number fraction of analyzed particles (*PF*) containing given chemical species measured by ALABAMA in the PASCAL snow samples as a function of the salinity classes (Sn). The selected species: sodium chloride (NaCl), non-sea-salt (nss) nitrate, nss sulfate, magnesium (Mg), levoglucosan and dicarboxylic acids, organic carbon (OC) and elemental carbon (EC). Chemical composition measured for particles in the 110-5000 nm diameter range.**



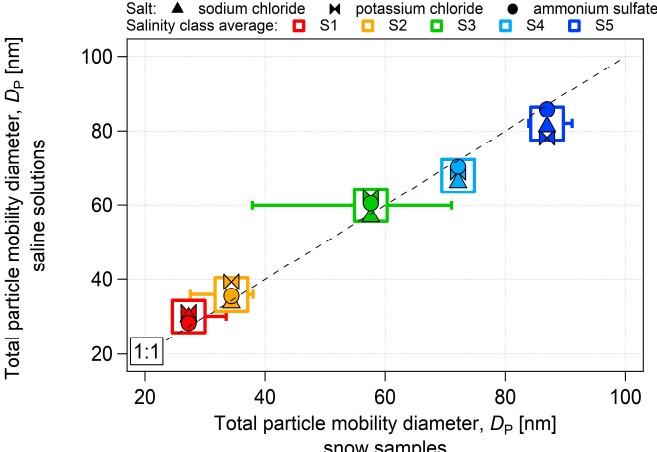

**Figure 4 Comparison of number concentration of aerosolized particles produced from inorganic salt solutions (sodium chloride, potassium chloride, ammonium sulfate) and from snow samples as function of salinity.**


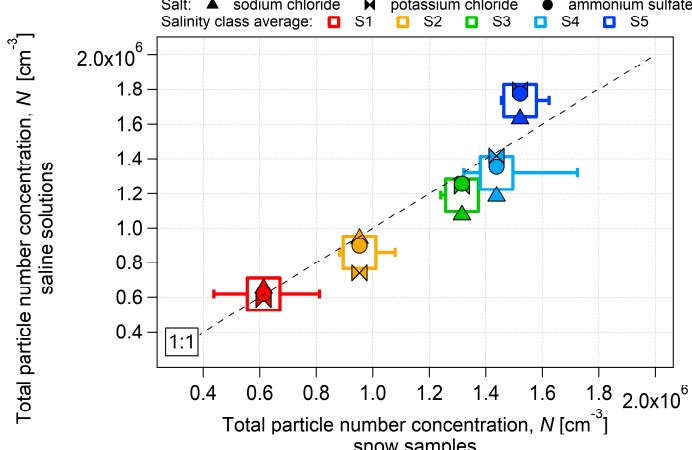


**Figure 5 Comparison of diameter of aerosolized particles produced from inorganic salt solutions (sodium chloride, potassium chloride, ammonium sulfate) and from snow samples as function of salinity. Diameter expressed as geometric mean diameter calculated from the number size distribution of aerosolized particles.**




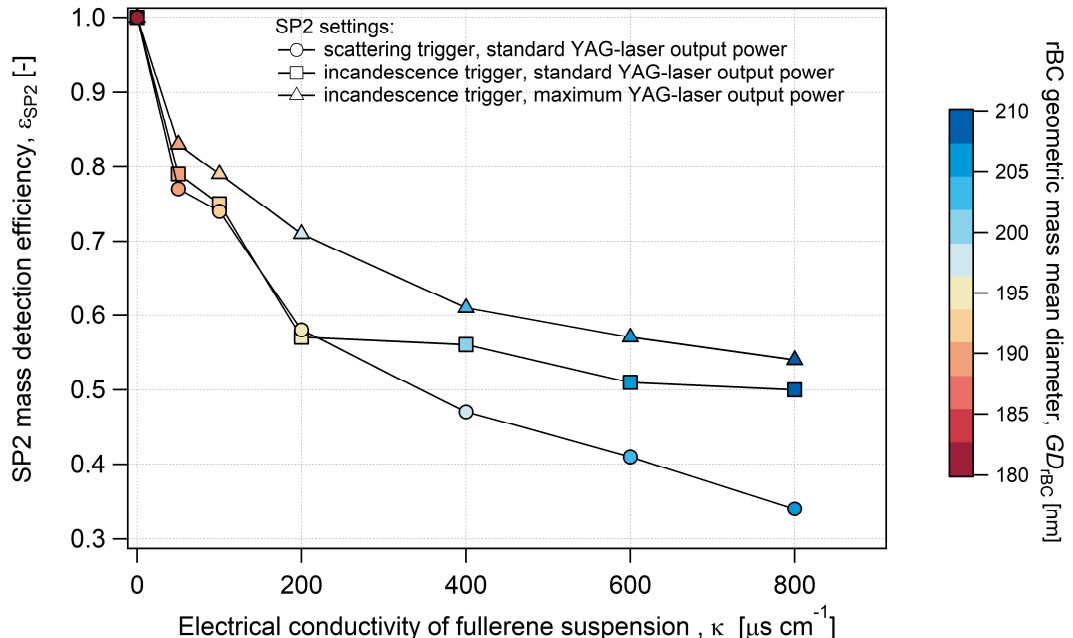

**Figure 6** Mass detection efficiency of the SP2 for different SP2 settings and increasing electrical conductivity. Data acquired from the analysis of a fullerene soot suspension at 10 μg l$^{-1}$.




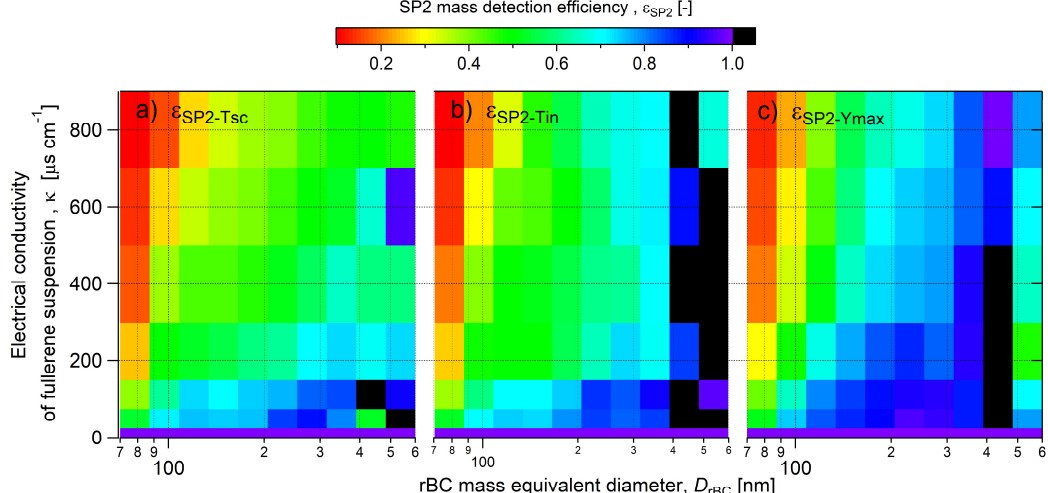

**Figure 7 rBC mass detection efficiency as function of $D_{rBC}$ for different SP2 settings. a) acquisistion triggered on the scattering detector with standard YAG-laser output power, $\varepsilon_{SP2-Tsc}$; b) acquisition triggered on the incandescence detector with standard YAG-laser output power $\varepsilon_{SP2-Tin}$; c) acquisition triggered on the incandescence detector with maximum YAG-laser output power, $\varepsilon_{SP2-Ymax}$. Results for fullerene soot suspensions with a concentration of 10 µg l$^{-1}$.**