# Peer review of "Technical note: sea salt interference with black carbon quantification in snow samples using the Single Particle Soot Photometer"

_Atmospheric Chemistry and Physics, 2021_

## Referee Comment (RC2)

**General comments:**

This is an important technical paper to accurately measure size-resolved BC mass and number concentrations in water samples including more non-BC particles (e.g., sea salt component). The authors investigated the impact of the sea salt on nebulization and rBC detection with SP2, by using an electrical conductivity ($\kappa$), to demonstrate that BC mass concentrations in water samples with more salinity are strongly affected by a SP2 mass detection efficiency. The paper is well organized and well written.

However, I have the following specific comments, minor comments, and technical corrections. Therefore, I recommend the major revision.

**Specific comments:**

1) Figure 2 and Table 1 are very interesting results. I am also interested in the bias of BC mass concentration in realistic surface snow samples (i.e., PASCAL snow samples). I suggest the investigation of the mass concentration of BC in the snow samples ($\mu$g/L) as a function of $\kappa$ and NaCl concentration, like Figure 2. This evaluation is very useful to accurately estimate the snow surface albedo, especially on the sea ice. In addition, how about the BC mass size distributions aerosolized from PASCAL snow samples for salinity classes ?

2) Authors investigated that the sea salt affects the operation of the SP2, by using a FS suspension of 10 $\mu$g/L which values are representative of BC mass concentration in central Arctic Ocean snow (3-15$\mu$g/g), measured by an ISSW method (Doherty et al., 2010). However, the concentration values might be overestimated due to an interference from coexisting non-BC solid particles (Schwarz et al., 2012; Mori et al., 2019). Some studies have shown the lower BC mass concentrations in surface snow and rainwater in the Arctic and Antarctic regions (< 5.0 $\mu$g/L) by using a nebulizer-SP2 technique (Kinase et al., 2020; Marquetto et al., 2020; Mori et al., 2019, 2020; Sinha et al., 2018). Therefore, it is also better to evaluate the salt impact on the rBC mass quantification for the lower FS mass concentrations (e.g., 1$\mu$g/L and 5$\mu$g/L), although the high $\kappa$ may cause the clogging of the SP2 aerosol jet.

3) Values of transport loss and efficiency are described in section 2.2 and 2.5.1, respectively. Although these values depend on microphysical properties (e.g., size, density, and shape) of aerosol particles, their transport pathway (e.g., length and thickness of the sampling line), and flow rate in the sampling lines, these detailed

calculations and assumptions are not shown in this manuscript. Please clarify how to estimate the transport loss or efficiency of the particles in the sampling line.

**Minor comments and Technical corrections:**

1) Abstract: This paper has evaluated the impact of sea salt, based on a measurement of an electrical conductivity ($\kappa$) in water samples, together with a nebulizer-SP2 technique. This point needs to be referred in the abstract.

2) Abstract L26: "We found strong correlations between both rBC mass concentration and rBC diameter with snow salinity".
rBC mass concentration → rBC number concentration ?

3) L77-79: Schwarz et al. (2012) and Mori et al. (2019) have also made a comparability with more traditional techniques (e.g. ISSW and TOT).

4) Table 1: I did not find the mass mixing ratio and geometric mean of the mass size distribution in this Table. In addition, some abbreviation parameters (e.g., $M_{rBC}$, $GD_{rBC}$, and $GD_p$) are not used in Table. Please add these parameters in this table.
"number and mass sized distribution ($GD_{rBC}$)" → "number and mass size distribution"

5) L118: Please add the measurement accuracy of the $\kappa$ values. The accuracy depends on the temperature in sample water ? Please clarify.

6) L121: Did you measure $\kappa$ in milliQ water? It is important to check that the $\kappa$ value is much lower than that for snow samples.

7) L127-129: Please add a nebulizer extraction efficiency used in this study.

8) L140-141: Do you make multiple charge correction and diffusion correction for the measurement of total aerosol number concentration? These corrections strongly control the concentration. Please clarify.

9) L143: Please specify "Different chemical species".

10) L145-152: Repeating some of what was stated on section 2.5. I suggest that this

sentence is moved to section 2.5.

11) L147-149: "In general, the transport losses for both lines were negligible (<3%) for particles in the 30-1000 nm diameter range, while slightly higher losses (below 7%) were calculated for particles smaller than 30 nm." Please add a reference.

12) Figure S1b and L203-204: Some studies suggested that sizes of BC particles in water sample increase by melt-refreezing cycles (Kinase et al., 2020). I think the different size distributions include this effect. Please clarify.

13) Figure S1 caption: "with the SMPS in the 14-680 nm diameter range nm" → "with the SMPS in the 14-680 nm diameter range"

14) L234: Figure number is different (Figure 4 → Figure 5)

15) L238: Figure number is different (Figure 5 → Figure 4)

16) Figure 4 caption: The figure caption does not explain the figure 4.

17) Figure 5 caption: The sentence related to a diameter is not needed.

18) L262-263: Please clarify the cause of the removal of solid salt obstruction.

19) L297-298: "varied 0.58 and 0.66" → "varied between 0.58 and 0.66" ?

20) Equation (1) and L313:
I think that the equation (1) is considered as the overall mass quantification efficiency of BC, if three efficiencies ($\varepsilon_{Neb}$, $\varepsilon_{SP2}$, and $\varepsilon_{Tm}$) are independent of the size and the $\varepsilon_{SP2}$ and $\varepsilon_{Tm}$ are 100% within 70–1000 nm range. Please clarify.

21) L340, L413: NaCl density is different in L340 and L413. In addition, please add a reference.

22) L381-382: Is the relationship between changes in BC sizes and $\kappa$ in FS samples similar to that for realistic PASCAL snow samples? Please add the mass concentration of BC and MMD in table 1.

23) L413-415: Mori et al. (2016) has estimated that the peak of the droplet size distribution, generated by the Marin-5 nebulizer, is 2-3 um, under condition of a gas flow rate of 1L/min and a liquid flow rate of 180 μL/min.

24) L417: from 59 nm at 50 μS cm$^{-1}$ to 197 nm at 800 μS cm$^{-1}$
→ "from 47 nm at 50 μS cm$^{-1}$ to 182 nm at 800 μS cm$^{-1}$"?

25) L421: Table S2 → Table S1

26) L427-430: Please add a new table showing the coating-rBC mass ratio, corresponding to table S1. This table is very useful to interpret the effect on the incandescence quenching.

27) Table 2: This table is not needed because the data have been already shown in Figure 6. In addition, this table is not referred in this manuscript.

References
1.  Kinase, T., Adachi, K., Oshima, N., Goto-Azuma, K., Ogawa-Tsukagawa, Y., Kondo, Y., et al (2020). Concentrations and size distributions of black carbon in the surface snow of eastern Antarctica in 2011. Journal of Geophysical Research: Atmospheres, 125, e2019JD030737. https://doi.org/10.1029/2019JD030737

2.  Marquetto, L., Kaspari, S., & Simões, J. C. (2020). Mass and number size distributions of rBC in snow and firn samples from Pine Island Glacier, West Antarctica. Earth and Space Science, 7, e2020EA001198. https://doi.org/10.1029/2020EA001198

3.  Mori, T., Moteki, N., Ohata, S., Koike, M., Goto-Azuma, K., Miyazaki, Y., & Kondo, Y. (2016). Improved technique for measuring the size distribution of black carbon particles in liquid water. Aerosol Science and Technology, 50(3), 242–254. https://doi.org/10.1080/02786826.2016.1147644

4.  Mori, T., Goto-Azuma, K., Kondo, Y., Ogawa-Tsukagawa, Y., Miura, K., Hirabayashi, M., et al. (2019). Black carbon and inorganic aerosols in Arctic snowpack. Journal of

Geophysical Research: Atmospheres, 124, 13,325–13,356. https://doi.org/10.1029/2019JD030623

5. Schwarz, J. P., Doherty, S. J., Li, F., Ruggiero, S. T., Tanner, C. E., Perring, A. E., et al. (2012). Assessing single particle soot photometer and integrating sphere/integrating sandwich spectrophotometer measurement techniques for quantifying black carbon concentration in snow. Atmospheric Measurement Techniques, 5(11), 2581–2592. https://doi.org/10.5194/amt-5-2581-2012

6. Sinha, P. R., Kondo, Y., Goto-Azuma, K., Tsukagawa, Y., Fukuda, K., Koike, M., et al. (2018). Seasonal progression of the deposition of black carbon by snowfall at Ny-Ålesund, Spitsbergen. Journal of Geophysical Research: Atmospheres, 123, 997–1016. https://doi.org/10.1002/2017JD028027

---

## Author Comment (AC1)

**ACP-2021-182: "Technical note: sea salt interference with black carbon quantification in snow samples using the Single Particle Soot Photometer"**

We would like to thank the referees for their detailed and constructive comments, which helped us to improve our manuscript. While the referee comments are given in **black bold,** our answers are given below in blue letters. Additionally, we added the changes made in the revised manuscript in **blue bold** letters.

**Answers of the authors to anonymous Reviewer#1**

**This paper is clearly written and presented. It presents important findings and useful information for anyone using an SP2 to measure rBC in snow or ice samples collected from environments where the samples could contain salt. My only comment is to note a few places where I suggest edits or where corrections are needed.**
Following the recommendation of the reviewer, we carefully revised the manuscript to improve its readability. The specific comments are addressed as follows. Note that an error was found in the section list. The "result section" now is labelled as Section 3 rather than 2.3.

**Overall, the paper could use a very light edit for English, but this is very minor.**
The paper underwent English polishing by a native speaker.

**Suggestion: In the title, capitalize Single Particle Soot Photometer, since it is a specific instrument.**
The text was modified following the reviewer's comment.

**Suggestion: Lines 90-92: "At the present time, the potential interference of sea salt during the analysis of rBC particles with the SP2 is not yet assessed." I would edit this to "Until now, the potential interference of sea salt during the analysis of rBC particles with the SP2 has not been assessed.", since with the publication of this paper it will have been.**
The text was modified following the reviewer's comment.

**Line 121 has an incomplete sentence.**
We apologize for this mistake. The incomplete sentence "Since κ increases with the concentration of ions in a solution" was removed.

**Lines 148-149 there is an errant carriage return splitting the word "higher"**
Corrected

**Line 157-158: another errant carriage return**
Corrected

**Figure 2: It would be helpful (but not necessary) for clarity if the left y-axis legend was in blue text and the right y-axis legend in red text**
Figure 2 was modified following reviewer#2 comment and now displays three separate panels.

**Figure 3: Consider changing the y-axis to log scale so that the variations in, e.g., EC, levoglucosan and organics can be seen better**
Unfortunately, due to the presence of zero values, changing the y-axis to log scale makes the variations of non-sea-salt species harder to read compared to a linear scale. For this reason, Figure 3 was not modified.

---

## Author Comment (AC2)

**ACP-2021-182: "Technical note: sea salt interference with black carbon quantification in snow samples using the Single Particle Soot Photometer"**

We would like to thank the referees for their detailed and constructive comments, which helped us to improve our manuscript. While the referee comments are given in **black bold,** our answers are given below in blue letters. Additionally, we added the changes we made in the revised manuscript in **blue bold** letters.

**Answers of the authors to anonymous Reviewer#2**

*GENERAL COMMENTS:*

**This is an important technical paper to accurately measure size-resolved BC mass and number concentrations in water samples including more non-BC particles (e.g., sea salt component). The authors investigated the impact of the sea salt on nebulization and rBC detection with SP2, by using an electrical conductivity ($\kappa$), to demonstrate that BC mass concentrations in water samples with more salinity are strongly affected by a SP2 mass detection efficiency. The paper is well organized and well written. However, I have the following specific comments, minor comments, and technical corrections. Therefore, I recommend the major revision.**
Following the recommendation of the reviewer, we carefully revised the manuscript to improve its readability. The specific and minor comments are addressed as follows. Note that an error was found in the section list. The "result section" now is labelled as Section 3 rather than 2.3.

*SPECIFIC COMMENTS:*

**1) Figure 2 and Table 1 are very interesting results. I am also interested in the bias of BC mass concentration in realistic surface snow samples (i.e., PASCAL snow samples). I suggest the investigation of the mass concentration of BC in the snow samples ($\mu$g/L) as a function of $\kappa$ and NaCl concentration, like Figure 2. This evaluation is very useful to accurately estimate the snow surface albedo, especially on the sea ice. In addition, how about the BC mass size distributions aerosolized from PASCAL snow samples for salinity classes ?**

>> We do agree that it is important to include the mass concentration and mass size distribution in Section 3.3.2 with the unique aim of supporting our technical assessment. Hence, Figure 2 was modified to accommodate $M_{rBC}$ as a function of $\kappa$. Similarly, Figure S1 now includes the rBC mass size distribution, while Table 1 now includes $M_{rBC}$ and the geometric mean of the mass size distribution. A new paragraph was added to Section 3.3.2 and it reads:
… **"The detected rBC mass mixing ratio ($M_{rBC}$) monotonically decreased with $\kappa$ from 8.05 ng g$^{-1}$ in S1 to 0.27 ng g$^{-1}$ in S5 (Erreur ! Source du renvoi introuvable.c). Although $M_{rBC}$ concentrations have already been observed in different locations and seasons across the Arctic (Sinha P. R. et al., 2017; Jacobi et al., 2019; Mori et al., 2019, 2020) and might be partially explained by BC redistribution during melting (Doherty et al., 2013); here the abrupt (factor 12) and sudden (48 hours) increase, observed between S3 and S1, of $M_{rBC}$ at the snow surface was never observed in the Arctic region and is not believed to be physical. The $M_{rBC}$ values presented here are corrected only for the mass nebulization efficiency (see more detail in Section Erreur ! Source du renvoi introuvable.). Similar to $GD_{rBC-N}$, the rBC geometric mean diameter calculated from the mass size distribution ($GD_{rBC-M}$), increased from 225 nm to 257 nm from S1 to S5 (Erreur ! Source du renvoi introuvable.). Considering that the melt-freeze cycle promotes agglomeration of BC particles (Schwarz et al., 2012; Kinase et al., 2020), the decrease of $GD_{rBC-M}$ at the melt onset (S1-S2) was unexpected. The respective rBC mass size distributions are shown in Erreur ! Source du renvoi introuvable.c."**…

>>Because of the sea-salt bias, the MrBC values are, most probably underestimated, thus not suited for snow albedo or forcing estimation, nor for model validation. An additional sentence explaining the limits of the $M_{rBC}$ results presented here was added at the end of Section 3.3.2 and it reads:

… **"As a consequence, the rBC properties presented in this section are prone to high error and might not be representative of natural processes only. This is particularly relevant for the most saline samples belonging to S3-S5"** …

**2) Authors investigated that the sea salt affects the operation of the SP2, by using a FS suspension of 10 µg/L which values are representative of BC mass concentration in central Arctic Ocean snow (3-15µg/g), measured by an ISSW method (Doherty et al., 2010). However, the concentration values might be overestimated due to an interference from coexisting non-BC solid particles (Schwarz et al., 2012; Mori et al., 2019). Some studies have shown the lower BC mass concentrations in surface snow and rainwater in the Arctic and Antarctic regions (< 5.0 µg/L) by using a nebulizer-SP2 technique (Kinase et al., 2020; Marquetto et al., 2020; Mori et al., 2019, 2020; Sinha et al., 2018). Therefore, it is also better to evaluate the salt impact on the rBC mass quantification for the lower FS mass concentrations (e.g., 1µg/L and 5µg/L), although the high κ may cause the clogging of the SP2 aerosol jet.**

>>We completely agree with the reviewer; it is important to evaluate the variability of $\varepsilon_{SP2}$ as a function of BC concentration. The first set of experiments were based on fullerene soot suspension at three concentrations: 1 µg l$^{-1}$ (FS01), 5 µg l$^{-1}$ (FS05), and 10 µg l$^{-1}$ (FS10). These experiments were performed by triggering the acquisition of the scattering signal. The variability of $\varepsilon_{SP2}$ as a function of κ is shown in Figure a for the various FS suspensions. $\varepsilon_{SP2}$ monotonically decreased with κ for all considered FS suspensions, while $GD_{MrBC}$ showed a common increasing trend with κ. Some differences are, however, observed in Figure a. FS10 and FS05 followed a very similar power decay as a function of κ. $\varepsilon_{SP2-FS05}$ slightly higher values above 200 µS cm$^{-1}$ than $\varepsilon_{SP2-FS10}$. $\varepsilon_{SP2-FS01}$ showed a more linear decrease as a function of κ. $\varepsilon_{SP2-FS01}$ was higher than $\varepsilon_{SP2-FS10}$ below approximately 400 µS cm$^{-1}$ and lower than $\varepsilon_{SP2-FS10}$ above approximately 600 µS cm$^{-1}$. Hence, Figure a does not indicate a univocal increase or decrease in $\varepsilon_{SP2}$ as a function of FS concentration. Due to the risks associated with clogging (under pressurization of the chamber) and its removal procedure (damaging and misalignment of the aerosol jet assembly), experiments with FS suspensions at 1 and 5 µg l$^{-1}$ were not repeated with incandescence triggering and max YAG-laser power. During the here presented experiments, clogging caused a drift in the sample flow rate and, potentially, a misalignment between the aerosol jet and the YAG-laser. We thus judged these results not to be robust enough for publication. The text of Section 3.2.2.2 was, however, partially reworked.

[Figure]

*Figure a Mass detection efficiency of the SP2 for FS suspensions at different concetration as function of electrical conductivity.*

**3 ) Values of transport loss and efficiency are described in section 2.2 and 2.5.1,respectively. Although these values depend on microphysical properties (e.g., size,density, and shape) of aerosol particles, their transport pathway (e.g., length andthickness of the sampling line), and flow rate in the sampling lines, these detailed calculations and assumptions are not shown in this manuscript. Please clarify how to estimate the transport loss or efficiency of the particles in the sampling line.**

>>More details on the losses calculations were added in the technical section 2.2 and in the result section 3.3.1. The new text in Section 2.2 reads:

… "Transport losses of aerosol particles were calculated using the Particle Loss Calculator software, which treats aerosol diffusion and sedimentation as well as turbulent inertial deposition and inertial deposition in bends and contractions of tubing (von der Weiden et al., 2009). The software was developed at the Max Planck Institute for Chemistry of Mainz (Germany) and is available for download at https://www.mpic.de/4230607/particle-loss-calculator-plc (last accessed 08 April 2021)." …

>>The new text in Section 3.3.1 now reads:

… "The transport losses were estimated for the SP2 sampling line, which was 30 cm long (distance from the Marin-5 exhaust) and composed of two different sections. The first section was 24 cm long with an internal diameter of 4.82 mm, an airflow rate of 1 L min$^{-1}$, and carried flow to SP2, SMPS, and ALABAMA. After a two-way flow splitter with a "Y" joint (model 1100; Brechtel, Hayward, USA), the SP2-specific sampling line was 6 cm long with an internal diameter of 1.8 mm, and an airflow rate of 0.12 L min$^{-1}$. No sharp bends or additional flow splitters were present along the second section. We assumed spherical particles with a void-free density of 1800 kg m$^{-3}$ (Moteki and Kondo, 2010). Losses of 2-5% and 2-16% were calculated for smaller (10-40 nm) and larger (1-5 μm) particles, respectively. The particle losses in the SP2 detection range (70-1000 nm of diameter) were estimated to remain well below 2%; hence not taken into account in forthcoming calculations." …

*MINOR COMMENTS AND TECHNICAL CORRECTIONS:*

**1) Abstract: This paper has evaluated the impact of sea salt, based on a measurement of an electrical conductivity (κ) in water samples, together with a nebulizer-SP2 technique. This point needs to be referred in the abstract.**
>>The related paragraph in the abstract now reads:
…" Based on the strong observational correlations between both rBC concentration and rBC diameter with snow salinity, we hypothesize a salt-induced matrix effect interfering with the SP2 analysis. This paper evaluates the impact of sea salt, based on the measurement of electrical conductivity (κ) in water samples, on rBC measurements made with a nebulizer-SP2 technique."…

**2) Abstract L26: "We found strong correlations between both rBC mass concentration and rBC diameter with snow salinity". rBC mass concentration → rBC number concentration?**
>>Corrected

**3) L77-79: Schwarz et al. (2012) and Mori et al. (2019) have also made a comparability with more traditional techniques (e.g. ISSW and TOT).**
>>The statement now reads:
…" the degree of comparability with more traditional techniques such as thermal-optical method and integrating sphere/integrating sandwich spectrophotometer is still variable (Schwarz et al., 2012; Lim et al., 2014; Mori et al., 2019)."…

**4) Table 1: I did not find the mass mixing ratio and geometric mean of the mass size distribution in this Table. In addition, some abbreviation parameters (e.g., *M*rBC, *GD*rBC, and *GD*p) are not used in Table. Please add these parameters in this table. "number and mass sized distribution (*GD*rBC)" → "number and mass size distribution"**
>>Table 1 and its legend were modified following the reviewer's suggestions.

**5) L118: Please add the measurement accuracy of the κ The accuracy depends on the temperature in sample water ? Please clarify.**

>>The accuracy of the probe does not change in the 5-80°C. However, the precision of our measurements did. We always measure κ in the water temperature range of 20-25°C. Considering that the electrical conductivity generally increases by 2% per 1°C increase (Hayashi, 2004), we estimate that our κ measurements had a precision of 10%. The text was modified accordingly.

**6) L121: Did you measure *κ* in milliQ water? It is important to check that the *κ* value is much lower than that for snow samples.**
>>The values of κ were checked for every experiment and never exceeded 1 μS cm$^{-1}$. This was clarified at the beginning of Section 2.2:
… **"The electrical conductivity of milliQ water newer exceeded 1 μS cm$^{-1}$, being considerably lower than all snow samples."** …

**7) L127-129: Please add a nebulizer extraction efficiency used in this study.**
>>The following statements was added:
… **"The mass nebulization efficiency varied between 0.58 and 0.66 (see Section** Erreur ! Source du renvoi introuvable. **for more details)."** …

**8) L140-141: Do you make multiple charge correction and diffusion correction for the measurement of total aerosol number concentration? These corrections strongly control the concentration. Please clarify.**
>>A new statement was added:
… **"Multiple charges and diffusion losses were corrected with the aerosol instrument manager software issued by DMT."** …

**9)L143: Please specify "Different chemical species".**
>>The statement now reads:
… **"Different chemical species, identified using characteristic marker ions of the mass spectra, include: sodium chloride, magnesium, non-sea-salt nitrate, non-sea-salt sulfate, organic carbon, levoglucosan-dicarboxilic acids, elemental carbon."** …

**10) L145-152: Repeating some of what was stated on section 2.5. I suggest that this sentence is moved to section 2.5.**
>>The sentence was moved to Section 2.5. Following Specific Comment 2, a description of the software is added instead at the end of Section 2.2.

**11) L147-149: "In general, the transport losses for both lines were negligible (<3%) for particles in the 30-1000 nm diameter range, while slightly higher losses (below 7%) were calculated for particles smaller than 30 nm." Please add a reference.**
>>More details on transport losses were added to the text following specific comment 2.

**12) Figure S1b and L203-204: Some studies suggested that sizes of BC particles in water sample increase by melt-refreezing cycles (Kinase et al., 2020). I think the different size distributions include this effect. Please clarify.**
>>This is a very interesting comment (see also answer to comment 22). S1-S2 samples were collected during and after the melting onset. While S3-S5 were collected before melting. Thus we would expect to observe larger particles in S1 compared to S5 samples. In reality, we observed the opposite trend. This is now underlined in Section 2.3.2:
… **"Similar to $GD_{rBC-N}$, the rBC geometric mean diameter calculated from the mass size distribution ($GD_{rBC-M}$), increased from 225 nm to 257 nm from S1 to S5 (**Erreur ! Source du renvoi introuvable.**). Considering that the melt-freeze cycle promotes agglomeration of BC particles (Schwarz et al., 2012; Kinase et al., 2020), the decrease of $GD_{rBC-M}$ at the melt onset (S1-S2) was unexpected."** …

**13) Figure S1 caption: "with the SMPS in the 14-680 nm diameter range nm" → "with the SMPS in the 14-680 nm diameter range"**
>>Corrected

**14) L234: Figure number is different (Figure 4 → Figure 5)**
>>Corrected

**15) L238: Figure number is different (Figure 5 → Figure 4)**
>>Corrected

**16) Figure 4 caption: The figure caption does not explain the figure 4.**
>>Corrected

**17) Figure 5 caption: The sentence related to a diameter is not needed.**
>>Corrected

**18) L262-263: Please clarify the cause of the removal of solid salt obstruction.**
>>The obstruction was caused by the accumulation of salt within the SP2-aerosol jet and caused a sudden drop in the flow rate. The full jet assembly was removed, the clogged section of the metal tubing was soaked in water and then sonicated for a very short period (less than 10 seconds). This allowed removing the salt obstruction. Note, Section 3.2.2.2 was completely rewritten to accommodate other comments.  The statement was modified and now reads:

… **"This finally formed a solid obstruction and caused the decreased sampling flow rate and the pressure at the detection chamber. To remove the salt obstruction, the full jet assembly was disconnected from the SP2, the clogged section of the metal tubing was first soaked in Milli-Q water, sonicated for a few seconds, and dried with compressed air."** …

**19) L297-298: "varied 0.58 and 0.66" → "varied between 0.58 and 0.66" ?**
>>Corrected

**20) Equation (1) and L313: I think that the equation (1) is considered as the overall mass quantification efficiency of BC, if three efficiencies ($\varepsilon$Neb, $\varepsilon$SP2, and $\varepsilon$Tm) are independent of the size and the $\varepsilon$SP2 and $\varepsilon$ Tm are 100% within 70–1000 nm range. Please clarify.**
>>The introduction paragraph of Section 3.3 was rearranged in several points in order to better explain $\varepsilon$Neb, $\varepsilon$SP2, and $\varepsilon$Tm. This part of the text now reads:

… **"$\varepsilon$ may be decomposed in three distinct contributions: the mass nebulization efficiency ($\varepsilon_{Neb}$), the transport efficiency ($\varepsilon_{Trn}$) and the mass detection efficiency of the SP2 ($\varepsilon_{SP2}$). $\varepsilon_{Neb}$ is the mass of rBC pumped into the nebulizer per unit time divided by the mass of rBC coming out of the nebulizer in aerosol form per unit time. Hence, $\varepsilon_{Neb}$ depends on the suspension efficiency of the liquid sample in liquid droplets and on the transport efficiency of the wet and dry aerosol within the nebulizer. $\varepsilon_{Trn}$ is the ratio of the mass of rBC pumped out the nebulizer per unit time to the mass of rBC reaching the SP2 sampling inlet per unit time. $\varepsilon_{Trn}$ is controlled by diffusion, impaction, and sedimentation losses, and depends on flow rate, particle size and the length, orientation, and bends of the sampling line. $\varepsilon_{SP2}$ is the mass of rBC in the aerosol form being introduced to the SP2 laser per unit time divided by the mass of rBC in aerosol form reported by the SP2 per unit of time. Considering typical atmospheric SP2 operation, $\varepsilon_{SP2}$ depends on the size distribution of rBC particles, the presence of other absorbing-refractory atmospheric components, and SP2 size detection range configuration (Schwarz et al., 2010). The ideal, but unrealistic, $\varepsilon$ of 1 indicates that 100% of rBC particles contained in the liquid sample are nebulized, transported, and finally detected by the SP2. In reality, the $\varepsilon$ values obtained for various soot standards suspended in pure water and nebulized with the Marin-5 nebulizer vary between 0.5 and 0.6 (Mori et al., 2016; Katich et al., 2017), while $\varepsilon$ of 0.75 was calculated by Lim et al. (2014) using an APEX-Q nebulizer. The $\varepsilon$ calculated in the present work for non-saline suspensions at different FS mass concentrations varied between 0.57 and 0.66. Our results are in good agreement with the previous studies, indicating good reproducibility of nebulization conditions during the laboratory experiments. It must be considered that past studies always assumed 100% $\varepsilon_{SP2}$ and $\varepsilon_{Trn}$ for rBC particles falling in the size detection range of the SP2. Therefore, $\varepsilon$ (calculated exactly as in this study according to Equation** Erreur ! Source du renvoi introuvable. **was usually addressed as nebulization efficiency. In this study, $\varepsilon$ decreased monotonically down to values between 0.33 and 0.21 for the most saline FS suspensions ($\kappa$ = 800 cm$^{-1}$). This decrease clearly indicated a salt induced bias on the mass quantification efficiency. Hence, the assumption of a 100% $\varepsilon_{SP2}$ and $\varepsilon_{Trn}$ might not be realistic for saline samples. The potential impact of salt presence on the nebulization and transport efficiency, and on the SP2 detection efficiency will be addressed in the following subsections."** …

**21) L340, L413: NaCl density is different in L340 and L413. In addition, please add a reference.**

>>Implemented

**22) L381-382: Is the relationship between changes in BC sizes and $\kappa$ in FS samples similar to that for realistic PASCAL snow samples? Please add the mass concentration of BC and MMD in table** 1.

>>Please, see also answer to Comment 12. The following sentence was added a bit later in the text (end of Section 3.3.2.2):

… **"These results are consistent with quenching of incandescence due to coatings, with stronger reductions in smaller and more thickly-coated rBC cores, and explains the size shift observed in the snow samples, where the $GD_{rBC-M}$ increases by approximately 14% from S1 to S5 samples (**Erreur ! Source du renvoi introuvable.**). A similar increase was observed in the laboratory test (SP2 operated at maximum YAG-laser power) where $GD_{rBC-M}$ increases from 183 nm at 0 µS cm$^{-1}$ to 211 nm at 800 µS cm$^{-1}$ (15% increase). This similarity suggests that incandescence quenching might have completely masked the diameter increase caused by particles agglomeration during melting-freezing cycles that may have existed in the snow samples (Schwarz et al., 2012; Kinase et al., 2020). "** …

**23) L413-415: Mori et al. (2016) has estimated that the peak of the droplet size distribution, generated by the Marin-5 nebulizer, is 2-3 um, under condition of a gas flow rate of 1L/min and a liquid flow rate of 180 µL/min.**

>>This is very interesting information. We used the same equation of Mori et al. (2016) to estimate a droplet diameter. Unfortunately, our and Mori's results are extremely different (see text below). This let me imagine that, despite logic, the equation of Mori is an oversimplification of the nebulization process and cannot be universally applied. The primary droplet diameter depends in fact on the nebulizer type, its inclination, its working pressure, the volume and temperature of the nebulization chamber, droplet coagulation, etc… Many studies and reviews in the field of inductively coupled plasma mass spectrometry (very briefly summarized in Section 2.5.1) addressed the issue. Nonetheless, Mori's approach is described in the text, which now reads:

… **"No direct measurements of the droplet size distribution are available for the Marine-5 nebulizer, but Mori et al. (2016) calculated a hypothetical droplet diameter of 2.5 µm based on the size distribution of total particles nebulized with the Marin-5 from an ammonium sulphate solution at a known concentration of 13.4 mg l$^{-1}$. By using the same equation, we calculated a droplet diameter of approximately 20 µm for a NaCl solution at a similar concentration of 13.1 mg l$^{-1}$ ($\kappa$=25 µS cm$^{-1}$, representative of S2). Considering the marked difference with the result of Mori et al. (2016), most probably due to an oversimplification of the nebulization process, we assumed a droplet diameter of 8 µm."** …

**24) L417: from 59 nm at 50 µS cm-1 to 197 nm at 800 µS cm-1→ "from 47 nm at 50 µS cm-1 to 182 nm at 800 µS cm-1"?**
>>These values were calculated for a BC diameter of 70nm and include in the text by mistake. Corrected.

**25) L421: Table S2 → Table S1**
>>Corrected

**26) L427-430: Please add a new table showing the coating-rBC mass ratio, corresponding to table S1. This table is very useful to interpret the effect on the incandescence quenching.**
>>Table S2 now contains the coating-rBC mass ratio, corresponding to table S1. In Section 2.5.2.2 the following statement was added:

… **"The theoretical coating-rBC mass ratio calculated from the coating thickness discussed above are presented in** Erreur ! Source du renvoi introuvable.**."** …

**27) Table 2: This table is not needed because the data have been already shown in Figure 6. In addition, this table is not referred in this manuscript.**
>>The table was removed.

**REFERENCES**

[revised manuscript text omitted]